# Multiplicative Weights Update with Constant Step-Size in Congestion Games: Convergence, Limit Cycles and Chaos

**Gerasimos Palaiopanos**[*]
SUTD
Singapore
gerasimosath@yahoo.com

**Ioannis Panageas**[†]
MIT
Cambridge, MA 02139
ioannis@csail.mit.edu

**Georgios Piliouras**[‡]
SUTD
Singapore
georgios@sutd.edu.sg

## Abstract

The Multiplicative Weights Update (MWU) method is a ubiquitous meta-algorithm that works as follows: A distribution is maintained on a certain set, and at each step the probability assigned to action $\gamma$ is multiplied by $(1 - \epsilon C(\gamma)) > 0$ where $C(\gamma)$ is the "cost" of action $\gamma$ and then rescaled to ensure that the new values form a distribution. We analyze MWU in congestion games where agents use *arbitrary admissible constants* as learning rates $\epsilon$ and prove convergence to *exact Nash equilibria*. Interestingly, this convergence result does not carry over to the nearly homologous MWU variant where at each step the probability assigned to action $\gamma$ is multiplied by $(1 - \epsilon)^{C(\gamma)}$ even for the simplest case of two-agent, two-strategy load balancing games, where such dynamics can provably lead to limit cycles or even chaotic behavior.

## 1 Introduction

The Multiplicative Weights Update (MWU) is a ubiquitous meta-algorithm with numerous applications in different fields [2]. It is particularly useful in game theory due to its regret-minimizing properties [24, 11]. It is typically introduced in two nearly identical variants, the one in which at each step the probability assigned to action $\gamma$ is multiplied by $(1 - \epsilon C(\gamma))$ and the one in which it is multiplied by $(1 - \epsilon)^{C(\gamma)}$ where $C(\gamma)$ is the cost of action $\gamma$. We will refer to the first as the linear variant, $\text{MWU}_\ell$, and the second as the exponential, $\text{MWU}_e$ (also known as Hedge). In the literature there is little distinction between these two variants as both carry the same advantageous regret-minimizing property. It is also well known that in order to achieve sublinear regret, the learning rate $\epsilon$ must be decreasing as time progresses. This constraint raises a natural question: Are there interesting classes of games where MWU behaves well without the need to fine-tune its learning rate?

A natural setting to test the learning behavior of MWU with constant learning rates $\epsilon$ is the well-studied class of congestion games. Unfortunately, even for the simplest instances of congestion games $\text{MWU}_e$ fails to converge to equilibria. For example, even in the simplest case of two balls two

---

[*]Gerasimos Palaiopanos would like to acknowledge a SUTD Presidential fellowship.

[†]Ioannis Panageas would like to acknowledge a MIT-SUTD postdoctoral fellowship. Part of this work was completed while Ioannis Panageas was a PhD student at Georgia Institute of Technology and a visiting scientist at the Simons Institute for the Theory of Computing.

[‡]Georgios Piliouras would like to acknowledge SUTD grant SRG ESD 2015 097, MOE AcRF Tier 2 Grant 2016-T2-1-170 and a NRF Fellowship. Part of this work was completed while Georgios Piliouras was a visiting scientist at the Simons Institute for the Theory of Computing.

bins games,[4] $\mathrm{MWU}_e$ with $\epsilon = 1 - e^{-10}$ is shown to converge to a limit cycle of period 2 for infinitely many initial conditions (Theorem 4.1). If the cost functions of the two edges are not identical then we create instances of two player load balancing games such that $\mathrm{MWU}_e$ has periodic orbits of length $k$ for all $k > 0$, as well as uncountable many initial conditions which never settle on any periodic orbit but instead exhibit an irregular behavior known as Li-Yorke chaos (Theorem 4.2, see Corollary 4.3).

The source of these problems is exactly the large, fixed learning rate $\epsilon$, *e.g.*, $\epsilon \approx 1$ for costs in $[0, 1]$. Intuitively, the key aspect of the problem can be captured by (simultaneous) best response dynamics. If both agents start from the same edge and best-respond simultaneously they will land on the second edge which now has a load of two. In the next step they will both jump back to the first edge and this motion will be continued perpetually. Naturally, $\mathrm{MWU}_e$ dynamics are considerably more intricate as they evolve over mixed strategies and allow for more complicated non-equilibrium behavior but the key insight is correct. Each agent has the right goal, decrease his own cost and hence the potential of the game, however, as they pursue this goal too aggressively they cancel each other's gains and lead to unpredictable non-converging behavior.

In a sense, the cautionary tales above agree with our intuition. Large, constant learning rates $\epsilon$ nullify the known performance guarantees of MWU. We *should* expect erratic behavior in such cases. The typical way to circumvent these problems is through careful monitoring and possibly successive halving of the $\epsilon$ parameter, a standard technique in the MWU literature. In this paper, we explore an alternative, cleaner, and surprisingly elegant solution to this problem. *We show that applying $\mathrm{MWU}_\ell$, the linear variant of MWU, suffices to guarantee convergence in all congestion games.*

**Our key contributions.** Our key result is the proof of convergence of $\mathrm{MWU}_\ell$ in congestion games. The main technical contribution is a proof that the potential of the mixed state is always strictly decreasing along any nontrivial trajectory (Theorem 3.1). This result holds for all congestion games, irrespective of the number of agents or the size, topology of the strategy sets. Moreover, each agent $i$ may be applying different learning rates $\epsilon_i$ which will be constant along the dynamics ($\epsilon_i$ *does not depend on the number of iterations $T$ of the dynamics and therefore is bounded away from zero as $T \to \infty$; this is not the case for most of the results in the literature*). The only restriction on the set of allowable learning rates $\epsilon_i$ is that for each agent the multiplicative factor $(1 - \epsilon_i C_i(\mathbf{s}))$ should be positive for all strategy outcomes $\mathbf{s}$.[5] Arguing convergence to equilibria for all initial conditions (Theorem 3.4) and further, convergence to Nash equilibria for all interior initial conditions (Theorem 3.8) follows. Proving that the potential always decreases (Theorem 3.1) hinges upon discovering a novel interpretation of MWU dynamics. Specifically, we show that the class of dynamical systems derived by applying $\mathrm{MWU}_\ell$ in congestion games is a special case of a convergent class of dynamical systems introduced by Baum and Eagon [5] (see Theorem 2.4). The most well known member of this class is the classic Baum-Welch algorithm, the standard instantiation of the Expectation-Maximization (EM) algorithm for hidden Markov models (HMM). Effectively, the proof of convergence of both these systems boils down to a proof of membership to the same class of Baum-Eagon systems (see section 2.3 for more details on these connections).

In the second part we provide simple congestion games where $\mathrm{MWU}_e$ provably fails to converge. The first main technical contribution of this section is proving convergence to a limit cycle, specifically a periodic orbit of length two, for the simplest case of two balls two bins games for infinitely many initial conditions (Theorem 4.1). Moreover, after normalizing costs to lie in $[0, 1]$, i.e. $c(x) = x/2$, we prove that almost all symmetric non-equilibrium initial conditions converge to a unique limit cycle when both agents use learning rate $\epsilon = 1 - e^{-10}$. In contrast, since $1 - \epsilon \cdot C(\mathbf{s}) \geq 1 - (1 - e^{-10})1 = e^{-10} > 0$, $\mathrm{MWU}_\ell$ successfully converges to equilibrium. In other words, for the *same learning rates*, $\mathrm{MWU}_e$ exhibits chaotic behavior whereas $\mathrm{MWU}_\ell$ converges to Nash equilibrium. Establishing chaotic behavior for the case of edges with different cost functions is rather straightforward in comparison (Theorem 4.2). The key step is to exploit symmetries in the system to reduce it to a single dimensional one and then establish the existence of a periodic orbit of length three. The existence of periodic orbits of any length as well as chaotic orbits then follows from the Li-Yorke theorem 2.3 [30] (see section 2.2 for background on chaos and dynamical systems). Finally, for any learning rate $1 > \epsilon > 0$, we construct $n$-player games so that $\mathrm{MWU}_e$ has chaotic behavior for uncountably many starting points.

**Related work and Extensions/Implications of our results.**

**Connections to learning in games and price of anarchy:** Several recent papers, e.g., [40, 22] focus on proving welfare guarantees of no-regret dynamics in games exploiting connections to (robust) price of anarchy literature [37] by establishing fast convergence of the time average behavior to (approximate) coarse correlate equilibria. Although these approaches are rather powerful they are not always applicable. For example, it is well known that when we consider the makespan (i.e. the load of the most congested machine) instead of the social/total cost there can be an exponential gap between the performance of coarse correlated equilibria and Nash equilibria. For example the price of anarchy for the makespan objective for $n$ balls $n$ bins games is $O(\log(n)/\log\log(n))$ whereas for the worst no regret algorithm it can be $\Omega(\sqrt{n})$ [9]. Moreover, even if we focus on the social cost, the price of anarchy guarantees do not carry over if we perform affine transformation to the cost functions (e.g. if there exist users of different tiers/types that the system designer wants to account for in a differential manner). In contrast, *our convergence results are robust to any affine cost transformation*. In fact, our results apply for all weighted potential games [32] (Remark 3.5).

**Connections to distributed computation and adversarial agent scheduling:** A rather realistic concern about results on learning in games has to do with their sensitivity to the ordering of the moves of the agent dynamics. For example, better-response dynamics in congestion games are guaranteed to converge only if in every round, exactly one agent deviates to a better strategy. A series of recent papers has established strong non-termination (cycling) results for large classes of bounded recall dynamics with a wide variety of interesting and timely applications: game theory, circuit design, social networks, routing and congestion control [26, 19, 34, 25]. In the case of games, these results translate to corollaries such as: "If there are two or more pure Nash equilibria in a game with unique best responses, then all bounded-recall self-independent dynamics[6] for which those equilibria are fixed points can fail to converge in asynchronous environments." Even the simplest 2 balls 2 bins game satisfies these properties (two pure Nash and unique best responses) which shows the strength of this impossibility result. In contrast, *our convergence result holds for any adversarial scheduling with the minimal fairness assumption that given any mixed state at least one agent who is not best responding eventually will be given the possibility to update their behavior*, answering open questions in [26, 25]. In fact, our convergence result is in a sense the strongest possible, no matter how many agents get to update their behavior (as long as one of them does) then the potential of the game will strictly decrease (Corollary 3.6).

**Connections to complexity theory:** Whereas the complexity of computing both mixed Nash equilibria in general games (PPAD-complete [17]) as well as the complexity of finding pure Nash equilibria in congestion games (PLS-complete [20]) have both been completely characterized and are thus unlikely to admit an efficient time algorithm, the complexity of computing mixed Nash equilibria in congestion games has withstood so far an exhaustive characterization. Naturally, it lies on the intersection of both PPAD and PLS, known as CLS [18]. Such an equilibrium can be found both via an end-of-line type of argument as well as a local search type of argument, but it is still not known if it is CLS-complete. Given the active interest for producing CLS-complete problems [16, 21] our constructive/convergence proof may help shed light on this open question.

**Chaos for arbitrary small learning rates $\epsilon$:** Although our example of chaotic behavior uses a very high learning rate $\epsilon = 1 - e^{-10}$, it should be noted that *for any learning rate $\epsilon$ (e.g. $\epsilon = e^{-10}$), as well as for any number of agents $n$, we can create congestion games with $n$ agents where $MWU_e$ exhibits chaotic behavior (Corollary 4.3).*

**Congestion/potential games:** Congestion games are amongst the most well known and thoroughly studied class of games. Proposed in [36] and isomorphic to potential games [32], they have been successfully employed in myriad modeling problems. Despite the numerous positive convergence results for concurrent dynamics in congestion games, e.g., [33, 23, 7, 1, 6, 28, 10, 13, 12, 31], we know of no prior work establishing such a *deterministic* convergence result of the day-to-day agent behavior to *exact* Nash equilibria for general atomic congestion games. MWU has also been studied in congestion games. In [29] randomized variants of the exponential version of the MWU are shown to converge w.h.p. to pure Nash equilibria as long as the learning rate $\epsilon$ is small enough. In contrast our positive results for linear $MWU_\ell$ hold deterministically and for all learning rates. Recently, [14] showed that if the Hedge algorithm is run with a suitably decreasing learning factor $\epsilon$, the sequence

of play converges to a Nash equilibrium with probability 1 (in the bandit case). The result and the techniques are orthogonal to ours, since we assume fixed learning rates.

**Non-convergent dynamics:** Outside the class of congestion games, there exist several negative results in the literature concerning the non-convergence of MWU and variants thereof. In particular, in [15] it was shown that the multiplicative updates algorithm fails to find the unique Nash equilibrium of the $3 \times 3$ Shapley game. Similar non-convergent results have been proven for perturbed zero-sum games [4], as well as for the continuous time version of MWU, the replicator dynamics [27, 35]. The possibility of applying Li-Yorke type arguments for MWU in congestion games with two agents was inspired by a remark in [3] for the case of continuum of agents. Our paper is the first to our knowledge where non-convergent MWU behavior in congestion games is formally proven capturing both limit cycles and chaos and we do so in the minimal case of two balls two bin games.

## 2 Preliminaries

**Notation.** We use boldface letters, e.g., $\mathbf{x}$, to denote column vectors (points). For a function $f : \mathbb{R}^m \to \mathbb{R}^m$, by $f^n$ we denote the composition of $f$ with itself $n$ times, namely $\underbrace{f \circ f \circ \cdots \circ f}_{n \text{ times}}$.

### 2.1 Congestion Games

A *congestion game* [36] is defined by the tuple $(\mathcal{N}; E; (S_i)_{i \in \mathcal{N}}; (c_e)_{e \in E})$ where $\mathcal{N}$ is the set of *agents*, $N = |\mathcal{N}|$, $E$ is a set of *resources* (also known as *edges* or *bins* or *facilities*) and each player $i$ has a set $S_i$ of subsets of $E$ ($S_i \subseteq 2^E$) and $|S_i| \geq 1$. Each strategy $s_i \in S_i$ is a set of edges and $c_e$ is a positive cost (latency) function associated with facility $e$. We use small greek characters like $\gamma, \delta$ to denote different strategies/paths. For a strategy profile $\mathbf{s} = (s_1, s_2, \ldots, s_N)$, the cost of player $i$ is given by $c_i(\mathbf{s}) = \sum_{e \in s_i} c_e(\ell_e(\mathbf{s}))$, where $\ell_e(\mathbf{s})$ is the number of players using $e$ in $\mathbf{s}$ (the load of edge $e$). The potential function is defined to be $\Phi(\mathbf{s}) = \sum_{e \in E} \sum_{j=1}^{\ell_e(\mathbf{s})} c_e(j)$.

For each $i \in \mathcal{N}$ and $\gamma \in S_i$, $p_{i\gamma}$ denotes the probability player $i$ chooses strategy $\gamma$. We denote by $\Delta(S_i) = \{\mathbf{p} \geq \mathbf{0} : \sum_{\gamma} p_{i\gamma} = 1\}$ the set of mixed (randomized) strategies of player $i$ and $\Delta = \times_i \Delta(S_i)$ the set of mixed strategies of all players. We use $c_{i\gamma} = \mathbb{E}_{\mathbf{s}_{-i} \sim \mathbf{p}_{-i}} c_i(\gamma, \mathbf{s}_{-i})$ to denote the expected cost of player $i$ given that he chooses strategy $\gamma$ and $\hat{c}_i = \sum_{\delta \in S_i} p_{i\delta} c_{i\delta}$ to denote his expected cost.

### 2.2 Dynamical Systems and Chaos

Let $\mathbf{x}^{(t+1)} = f(\mathbf{x}^{(t)})$ be a *discrete time* dynamical system with update rule $f : \mathbb{R}^m \to \mathbb{R}^m$. The point $\mathbf{z}$ is called a *fixed point* of $f$ if $f(\mathbf{z}) = \mathbf{z}$. A sequence $(f^t(\mathbf{x}^{(0)}))_{t \in \mathbb{N}}$ is called a *trajectory* or *orbit* of the dynamics with $x^{(0)}$ as starting point. A common technique to show that a dynamical system converges to a fixed point is to construct a function $P : \mathbb{R}^m \to \mathbb{R}$ such that $P(f(\mathbf{x})) > P(\mathbf{x})$ unless $\mathbf{x}$ is a fixed point. We call $P$ a *Lyapunov* or *potential* function.

**Definition 2.1.** $C = \{\mathbf{z}_1, \ldots, \mathbf{z}_k\}$ *is called a periodic orbit of length* $k$ *if* $\mathbf{z}_{i+1} = f(\mathbf{z}_i)$ *for* $1 \leq i \leq k - 1$ *and* $f(\mathbf{z}_k) = \mathbf{z}_1$. *Each point* $\mathbf{z}_1, \ldots, \mathbf{z}_k$ *is called periodic point of period* $k$. *If the dynamics converges to some periodic orbit, we also use the term limit cycle.*

Some dynamical systems converge and their behavior can be fully understood and some others have strange, *chaotic* behavior. There are many different definitions for what chaotic behavior and chaos means. In this paper we follow the definition of chaos by Li and Yorke. Let us first give the definition of a scrambled set. Given a dynamical system with update rule $f$, a pair $x$ and $y$ is called "scrambled" if $\lim_{n \to \infty} \inf |f^n(x) - f^n(y)| = 0$ (the trajectories get arbitrarily close) and also $\lim_{n \to \infty} \sup |f^n(x) - f^n(y)| > 0$ (the trajectories move apart). A set $S$ is called "scrambled" if $\forall x, y \in S$, the pair is "scrambled".

**Definition 2.2** (Li and Yorke)**.** *A discrete time dynamical system with update rule* $f$, $f : X \to X$ *continuous on a compact set* $X \subset \mathbb{R}$ *is called chaotic if (a) for each* $k \in \mathbb{Z}^+$, *there exists a periodic point* $p \in X$ *of period* $k$ *and (b) there is an uncountably infinite set* $S \subseteq X$ *that is "scrambled".*

Li and Yorke proved the following theorem [30] (there is another theorem of similar flavor due to Sharkovskii [38]):

**Theorem 2.3** (Period three implies chaos). *Let $J$ be an interval and let $F : J \to J$ be continuous. Assume there is a point $a \in J$ for which the points $b = F(a), c = F^2(a)$ and $d = F^3(a)$, satisfy*

$$d \leq a < b < c \text{ (or } d \geq a > b > c).$$

*Then*

1. *For every $k = 1, 2, \dots$ there is a periodic point in $J$ having period $k$.*

2. *There is an uncountable set $S \subset J$ (containing no periodic points), which satisfies the following conditions:*

   - *For every $p, q \in S$ with $p \neq q$,*

     $$\lim_{n \to \infty} \sup |F^n(p) - F^n(q)| > 0 \text{ and } \lim_{n \to \infty} \inf |F^n(p) - F^n(q)| = 0.$$

   - *For every point $p \in S$ and periodic point $q \in J$,*

     $$\lim_{n \to \infty} \sup |F^n(p) - F^n(q)| > 0.$$

*Notice that if there is a periodic point with period $3$, then the hypothesis of the theorem will be satisfied.*

### 2.3 Baum-Eagon Inequality, Baum-Welch and EM

We start this subsection by stating the Baum-Eagon inequality. This inequality will be used to show that MWU$_\ell$ converges to fixed points and more specifically Nash equilibria for congestion games.

**Theorem 2.4** (Baum-Eagon inequality [5]). *Let $P(\mathbf{x}) = P(\{x_{ij}\})$ be a polynomial with nonnegative coefficients homogeneous of degree $d$ in its variables $\{x_{ij}\}$. Let $\mathbf{x} = \{x_{ij}\}$ be any point of the domain $D : x_{ij} \geq 0, \sum_{j=1}^{q_i} x_{ij} = 1, i = 1, 2, ..., p, j = 1, 2, ..., q_i$. For $\mathbf{x} = \{x_{ij}\} \in D$ let $\Im(\mathbf{x}) = \Im\{x_{ij}\}$ denote the point of $D$ whose i, j coordinate is*

$$\Im(\mathbf{x})_{ij} = \left( x_{ij} \frac{\partial P}{\partial x_{ij}} \bigg|_{(\mathbf{x})} \right) \bigg/ \sum_{j'=1}^{q_i} x_{ij'} \frac{\partial P}{\partial x_{ij'}} \bigg|_{(\mathbf{x})}$$

*Then $P(\Im(\mathbf{x})) > P(\mathbf{x})$ unless $\Im(\mathbf{x}) = \mathbf{x}$.*

The Baum-Welch algorithm is a classic technique used to find the unknown parameters of a hidden Markov model (HMM). A HMM describes the joint probability of a collection of "hidden" and observed discrete random variables. It relies on the assumption that the $i$-th hidden variable given the $(i-1)$-th hidden variable is independent of previous hidden variables, and the current observation variables depend only on the current hidden state. The Baum-Welch algorithm uses the well known EM algorithm to find the maximum likelihood estimate of the parameters of a hidden Markov model given a set of observed feature vectors. More detailed exposition of these ideas can be found here [8]. The probability of making a specific time series of observations of length $T$ can be shown to be a homogeneous polynomial $P$ of degree $T$ with nonnegative (integer) coefficients of the model parameters. Baum-Welch algorithm is homologous to the iterative process derived by applying the Baum-Eagon theorem to polynomial $P$ [5, 41].

In a nutshell, both Baum-Welch and MWU$_\ell$ in congestion games are special cases of the Baum-Eagon iterative process (for different polynomials $P$).

### 2.4 Multiplicative Weights Update

In this section, we describe the MWU dynamics (both the linear MWU$_\ell$, and the exponential MWU$_e$ variants) applied in congestion games. The update rule (function) $\xi : \Delta \to \Delta$ (where $\mathbf{p}(t+1) = \xi(\mathbf{p}(t))$) for the linear variant MWU$_\ell$ is as follows:

$$p_{i\gamma}(t+1) = (\xi(\mathbf{p}(t)))_{i\gamma} = p_{i\gamma}(t) \frac{1 - \epsilon_i c_{i\gamma}(t)}{1 - \epsilon_i \hat{c}_i(t)}, \quad \forall i \in \mathcal{N}, \forall \gamma \in S_i, \tag{1}$$

where $\epsilon_i$ is a constant (can depend on player $i$ but not on $\mathbf{p}$) so that both enumerator and denominator of the fraction in (1) are positive (and thus the fraction is well defined). Under the assumption that $1/\epsilon_i > \frac{1}{\beta} \stackrel{\text{def}}{=} \sup_{i,\mathbf{p}\in\Delta,\gamma\in S_i}\{c_{i\gamma}\}$, it follows that $1/\epsilon_i > c_{i\gamma}$ for all $i,\gamma$ and hence $1/\epsilon_i > \hat{c}_i$.

The update rule (function) $\eta : \Delta \rightarrow \Delta$ (where $\mathbf{p}(t+1) = \eta(\mathbf{p}(t))$) for the exponential variant $\text{MWU}_e$ is as follows:

$$p_{i\gamma}(t+1) = (\eta(\mathbf{p}(t)))_{i\gamma} = p_{i\gamma}(t) \frac{(1-\epsilon_i)^{c_{i\gamma}(t)}}{\sum_{\gamma'\in S_i} p_{i\gamma'}(t)(1-\epsilon_i)^{c_{i\gamma'}(t)}}, \quad \forall i \in \mathcal{N}, \forall \gamma \in S_i, \quad (2)$$

where $\epsilon_i < 1$ is a constant (can depend on player $i$ but not on $\mathbf{p}$). Note that $\epsilon_i$ can be small when the number of agents $N$ is large enough.

**Remark 2.5.** *Observe that $\Delta$ is invariant under the discrete dynamics (1), (2) defined above. If $p_{i\gamma} = 0$ then $p_{i\gamma}$ remains zero, and if it is positive, it remains positive (both numerator and denominator are positive) and also is true that $\sum_{\gamma\in S_i} p_{i\gamma} = 1$ for all agents $i$. A point $\mathbf{p}^*$ is called a fixed point if it stays invariant under the update rule of the dynamics, namely $\xi(\mathbf{p}^*) = \mathbf{p}^*$ or $\eta(\mathbf{p}^*) = \mathbf{p}^*$. A point $\mathbf{p}^*$ is a fixed point of (1), (2) if for all $i,\gamma$ with $p_{i\gamma}^* > 0$ we have that $c_{i\gamma} = \hat{c}_i$. To see why, observe that if $p_{i\gamma}^*, p_{i\gamma'}^* > 0$, then $c_{i\gamma} = c_{i\gamma'}$ and thus $c_{i\gamma} = \hat{c}_i$. We conclude that the set of fixed points of both dynamics (1), (2) coincide and are supersets of the set of Nash equilibria of the corresponding congestion game.*

## 3 Convergence of MWU$_\ell$ to Nash Equilibria

We first prove that MWU$_\ell$ (1) converges to fixed points[7]. Technically, we establish that function $\Psi \stackrel{\text{def}}{=} \mathbb{E}_{\mathbf{s}\sim\mathbf{p}}[\Phi(\mathbf{s})]$ is strictly decreasing along any nontrivial (i.e. nonequilibrium) trajectory, where $\Phi$ is the potential function of the congestion game as defined in Section 2. Formally we show the following theorem:

**Theorem 3.1** ($\Psi$ is decreasing)**.** *Function $\Psi$ is decreasing w.r.t. time, i.e., $\Psi(\mathbf{p}(t+1)) \leq \Psi(\mathbf{p}(t))$ where equality $\Psi(\mathbf{p}(t+1)) = \Psi(\mathbf{p}(t))$ holds **only** at fixed points.*

We define the function

$$Q(\mathbf{p}) \stackrel{\text{def}}{=} \underbrace{\sum_{i\in\mathcal{N}} \left( (1/\epsilon_i - 1/\beta) \cdot \sum_{\gamma\in S_i} p_{i\gamma} \right) + 1/\beta \cdot \prod_{i\in\mathcal{N}} \left( \sum_{\gamma\in S_i} p_{i\gamma} \right)}_{\text{constant term}} - \Psi(\mathbf{p}), \quad (3)$$

and show that $Q(\mathbf{p})$ is strictly increasing w.r.t time, unless $\mathbf{p}$ is a fixed point. Observe that $\sum_{\gamma\in S_i} p_{i\gamma} = 1$ since $\mathbf{p}$ lies in $\Delta$, but we include this terms in $Q$ for technical reasons that will be made clear later in the section. By showing that $Q$ is increasing with time, Theorem 3.1 trivially follows since $Q = const - \Psi$ where $const = \sum_{i\in\mathcal{N}} 1/\epsilon_i - 1/\beta(N-1)$. To show that $Q(\mathbf{p})$ is strictly increasing w.r.t time, unless $\mathbf{p}$ is a fixed point, we use a generalization of an inequality by Baum and Eagon [5] on function $Q$.

**Corollary 3.2** (Generalization of Baum-Eagon)**.** *Theorem 2.4 holds even if $P$ is non-homogeneous.*

We want to apply Corollary 3.2 on $Q$. To do so, it suffices to show that $Q(\mathbf{p})$ is a polynomial with nonnegative coefficients.

**Lemma 3.3.** *$Q(\mathbf{p})$ is a polynomial with respect to $p_{i\gamma}$ and has nonnegative coefficients.*

Using Lemma 3.3 and Corollary 3.2 we show the following:

**Theorem 3.4.** *Let $Q$ be the function defined in (3). Let also $\mathbf{p}(t) \in \Delta$ be the point MWU$_\ell$ (1) outputs at time $t$ with update rule $\xi$. It holds that $Q(\mathbf{p}(t+1)) \stackrel{\text{def}}{=} Q(\xi(\mathbf{p}(t))) > Q(\mathbf{p}(t))$ unless $\xi(\mathbf{p}(t)) = \mathbf{p}(t)$ (fixed point). Namely $Q$ is strictly increasing with respect to the number of iterations $t$ unless MWU$_\ell$ is at a fixed point.*

**Remark 3.5** (Weighted potential games). *A congestion game is a potential game because if a player deviates, the difference he experiences in his cost is exactly captured by the deviation of the global (same for all players) function $\Phi = \sum_{e \in E} \sum_{j=1}^{\ell_e(\mathbf{s})} c_e(j)$. In a weighted potential game, it holds that $c_i(s_i, \mathbf{s}_{-i}) - c_i(s_i', \mathbf{s}_{-i}) = w_i(\Phi(s_i, \mathbf{s}_{-i}) - \Phi(s_i', \mathbf{s}_{-i}))$, where $w_i$ is some constant not necessarily 1 (as in the potential games case) and vector $\mathbf{s}_{-i}$ captures the strategies of all players but $i$. It is not hard to see that Lemma 3.3 and thus Theorems 3.4 and 3.1 hold in this particular class of games (which is a generalization of congestion games), and so do the rest of the theorems of the section. Effectively, in terms of the weighted potential games analysis, it is possible to reduce it to the standard potential games analysis as follows: Consider the system with learning rates $\epsilon_i$ and cost functions $w_i c_i$ so that the game with cost functions $c_i$ is a potential game. The only necessary condition that we ask of this system is that $\epsilon_i w_i c_i(\mathbf{s}) < 1$ for all $i$ (as in the standard case) so that the enumerators/denominators are positive.*

By reduction, we can show that for every round $T$, even if a subset (that depends on the round $T$) of the players update their strategy according to $\mathrm{MWU}_\ell$ and the rest remain fixed, the potential still decreases.

**Corollary 3.6** (Any subset). *Assume that at time $t$ we partition the players in two sets $S_t, S_t'$ so that we allow only players in $S_t$ to apply $\mathrm{MWU}_\ell$ dynamics, whereas the players in $S_t'$ remain fixed. It holds that the expected potential function of the game at time $t$ decreases.*

As stated earlier in the section, if $Q(\mathbf{p}(t))$ is strictly increasing with respect to time $t$ unless $\mathbf{p}(t)$ is a fixed point, it follows that the expected potential function $\Psi(\mathbf{p}(t)) = const - Q(\mathbf{p}(t))$ is strictly decreasing unless $\mathbf{p}(t)$ is a fixed point and Theorem 3.1 is proved. Moreover, we can derive the fact that our dynamics converges to fixed points as a corollary of Theorem 3.1.

**Theorem 3.7** (Convergence to fixed points). *$\mathrm{MWU}_\ell$ dynamics (1) converges to fixed points.*

We conclude the section by strengthening the convergence result (*i.e.,* Theorem 3.7). We show that if the initial distribution $\mathbf{p}$ is in the interior of $\Delta$ then we have convergence to Nash equilibria.

**Theorem 3.8** (Convergence to Nash equilibria). *Assume that the fixed points of (1) are isolated. Let $\mathbf{p}(0)$ be a point in the interior of $\Delta$. It follows that $\lim_{t \to \infty} \mathbf{p}(t) = \mathbf{p}^*$ is a Nash equilibrium.*

*Proof.* We showed in Theorem 3.7 that $\mathrm{MWU}_\ell$ dynamics (1) converges, hence $\lim_{t \to \infty} \mathbf{p}(t)$ exists (under the assumption that the fixed points are isolated) and is equal to a fixed point of the dynamics $\mathbf{p}^*$. Also it is clear from the dynamics that $\Delta$ is invariant, *i.e.,* $\sum_{\delta \in S_j} p_{j\delta}(t) = 1, p_{j\delta}(t) > 0$ for all $j$ and $t \geq 0$ since $\mathbf{p}(0)$ is in the interior of $\Delta$.

Assume that $\mathbf{p}^*$ is not a Nash equilibrium, then there exists a player $i$ and a strategy $\gamma \in S_i$ so that $c_{i\gamma}(\mathbf{p}^*) < \hat{c}_i(\mathbf{p}^*)$ (on mixed strategies $\mathbf{p}^*$) and $p_{i\gamma}^* = 0$. Fix a $\zeta > 0$ and let $U_\zeta = \{\mathbf{p} : c_{i\gamma}(\mathbf{p}) < \hat{c}_i(\mathbf{p}) - \zeta\}$. By continuity we have that $U_\zeta$ is open. It is also true that $\mathbf{p}^* \in U_\zeta$ for $\zeta$ small enough.

Since $\mathbf{p}(t)$ converges to $\mathbf{p}^*$ as $t \to \infty$, there exists a time $t_0$ so that for all $t' \geq t_0$ we have that $\mathbf{p}(t') \in U_\zeta$. However, from $\mathrm{MWU}_\ell$ dynamics (1) we get that if $\mathbf{p}(t') \in U_\zeta$ then $1 - \epsilon_i c_{i\gamma}(t') > 1 - \epsilon_i \hat{c}_i(t')$ and hence $p_{i\gamma}(t'+1) = p_{i\gamma}(t') \frac{1 - \epsilon_i c_{i\gamma}(t')}{1 - \epsilon_i \hat{c}_i(t')} \geq p_{i\gamma}(t') > 0$, *i.e.,* $p_{i\gamma}(t')$ is positive and increasing with $t' \geq t_0$. We reached a contradiction since $p_{i\gamma}(t) \to p_{i\gamma}^* = 0$, thus $\mathbf{p}^*$ is a Nash equilibrium. $\square$

# 4 Non-Convergence of $\mathrm{MWU}_e$: Limit Cycle and Chaos

We consider a symmetric two agent congestion game with two edges $e_1, e_2$. Both agents have the same two available strategies $\gamma_1 = \{e_1\}$ and $\gamma_2 = \{e_2\}$. We denote $x, y$ the probability that the first and the second agent respectively choose strategy $\gamma_1$.

For the first example, we assume that $c_{e_1}(l) = \frac{1}{2} \cdot l$ and $c_{e_2}(l) = \frac{1}{2} \cdot l$. Computing the expected costs we get that $c_{1\gamma_1} = \frac{1+y}{2}, c_{1\gamma_2} = \frac{2-y}{2}, c_{2\gamma_1} = \frac{1+x}{2}, c_{2\gamma_2} = \frac{2-x}{2}$. $\mathrm{MWU}_e$ then becomes $x_{t+1} = x_t \frac{(1-\epsilon_1)^{\frac{(y_t+1)}{2}}}{x_t(1-\epsilon_1)^{\frac{y_t+1}{2}} + (1-x_t)(1-\epsilon_1)^{\frac{2-y_t}{2}}}$ (first player) and $y_{t+1} = y_t \frac{(1-\epsilon_2)^{\frac{x_t+1}{2}}}{y_t(1-\epsilon_2)^{\frac{x_t+1}{2}} + (1-y_t)(1-\epsilon_2)^{\frac{2-x_t}{2}}}$ (second player). We assume that $\epsilon_1 = \epsilon_2$ and also that $x_0 = y_0$ (players start with the same mixed

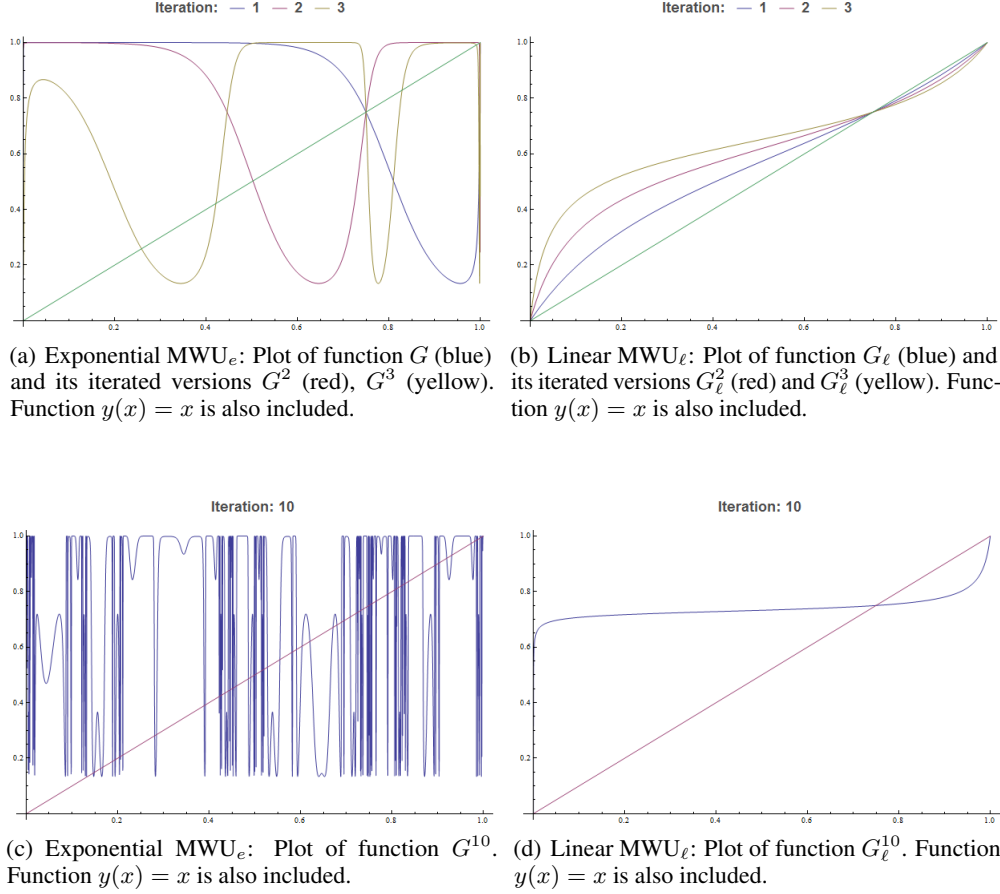

(a) Exponential $MWU_e$: Plot of function $G$ (blue) and its iterated versions $G^2$ (red), $G^3$ (yellow). Function $y(x) = x$ is also included.

(b) Linear $MWU_\ell$: Plot of function $G_\ell$ (blue) and its iterated versions $G_\ell^2$ (red) and $G_\ell^3$ (yellow). Function $y(x) = x$ is also included.

(c) Exponential $MWU_e$: Plot of function $G^{10}$. Function $y(x) = x$ is also included.

(d) Linear $MWU_\ell$: Plot of function $G_\ell^{10}$. Function $y(x) = x$ is also included.

Figure 1: We compare and contrast $MWU_e$ (left) and $MWU_\ell$ (right) in the same two agent two strategy/edges congestion game with $c_{e_1}(l) = \frac{1}{4} \cdot l$ and $c_{e_2}(l) = \frac{1.4}{4} \cdot l$ and same learning rate $\epsilon = 1 - e^{-40}$. $MWU_e$ exhibits sensitivity to initial conditions whereas $MWU_\ell$ equilibrates. Function $y(x) = x$ is also included in the graphs to help identify fixed points and periodic points.

strategy. Due to symmetry, it follows that $x_t = y_t$ for all $t \in \mathbb{N}$, thus it suffices to keep track only of one variable (we have reduced the number of variables of the update rule of the dynamics to one) and the dynamics becomes $x_{t+1} = x_t \frac{(1-\epsilon)^{\frac{x_t+1}{2}}}{x_t(1-\epsilon)^{\frac{x_t+1}{2}} + (1-x_t)(1-\epsilon)^{\frac{2-x_t}{2}}}$. Finally, we choose $\epsilon = 1 - e^{-10}$ and we get

$$x_{t+1} = H(x_t) = x_t \frac{e^{-5(x_t+1)}}{x_t e^{-5(x_t+1)} + (1-x_t)e^{-5(2-x_t)}},$$

i.e., we denote $H(x) = \frac{xe^{-5(x+1)}}{xe^{-5(x+1)} + (1-x)e^{-5(2-x)}}$.

For the second example, we assume that $c_{e_1}(l) = \frac{1}{4} \cdot l$ and $c_{e_2}(l) = \frac{1.4}{4} \cdot l$. Computing the expected costs we get that $c_{1\gamma_1} = \frac{1+y}{4}$, $c_{1\gamma_2} = \frac{1.4(2-y)}{4}$, $c_{2\gamma_1} = \frac{1+x}{4}$, $c_{2\gamma_2} = \frac{1.4(2-x)}{4}$.
$MWU_e$ then becomes $x_{t+1} = x_t \frac{(1-\epsilon_1)^{\frac{(y_t+1)}{4}}}{x_t(1-\epsilon_1)^{\frac{y_t+1}{4}} + (1-x_t)(1-\epsilon_1)^{\frac{1.4(2-y_t)}{4}}}$ (first player) and $y_{t+1} =$
$y_t \frac{(1-\epsilon_2)^{\frac{x_t+1}{4}}}{y_t(1-\epsilon_2)^{\frac{x_t+1}{4}} + (1-y_t)(1-\epsilon_2)^{\frac{1.4(2-x_t)}{4}}}$ (second player). We assume that $\epsilon_1 = \epsilon_2$ and also that $x_0 = y_0$ (players start with the same mixed strategy. Similarly, due to symmetry, it follows that $x_t = y_t$ for all $t \in \mathbb{N}$, thus it suffices to keep track only of one variable and the dynamics becomes

$$x_{t+1} = x_t \frac{(1-\epsilon)^{\frac{x_t+1}{4}}}{x_t(1-\epsilon)^{\frac{x_t+1}{4}}+(1-x_t)(1-\epsilon)^{\frac{1.4(2-x_t)}{4}}}.$$ Finally, we choose $\epsilon = 1 - e^{-40}$ and we get

$$x_{t+1} = G(x_t) = x_t \frac{e^{-10(x_t+1)}}{x_t e^{-10(x_t+1)} + (1-x_t)e^{-14(2-x_t)}},$$

i.e., we denote $G(x) = \frac{xe^{-10(x+1)}}{xe^{-10(x+1)}+(1-x)e^{-14(2-x)}}$.

We show the following three statements, the proofs of which can be found in the full version.

**Theorem 4.1.** *For all but a measure zero set $S$ of $x \in (0,1)$ we get that $\lim_{t\to\infty} H^{2t}(x) = \rho_1$ or $\rho_2$. Moreover, $H(\rho_1) = \rho_2$ and $H(\rho_2) = \rho_1$, i.e., $\{\rho_1, \rho_2\}$ is a periodic orbit. Thus, all but a measure zero set $S$ of initial conditions converge to the limit cycle $\{\rho_1, \rho_2\}$. Finally, the initial points in $S$ converge to the equilibrium $\frac{1}{2}$.*

**Theorem 4.2.** *There exist two player two strategy symmetric congestion games such that $MWU_e$ has periodic orbits of length $n$ for any natural number $n > 0$ and as well as an uncountably infinite set of "scrambled" initial conditions (Li-Yorke chaos).*

Using Theorem 4.2, we conclude with the following corollary.

**Corollary 4.3.** *For any $1 > \epsilon > 0$ and $n$, there exists a $n$-player congestion game $G(\epsilon)$ (depending on $\epsilon$) so that $MWU_e$ dynamics exhibits Li-Yorke chaos for uncountably many starting points.*

## 5 Conclusion and Future Work

We have analyzed $MWU_\ell$ in congestion games where agents use *arbitrary admissible constants* as learning rates $\epsilon$ and showed convergence to *exact Nash equilibria*. We have also shown that this result is not true for the nearly homologous exponential variant $MWU_e$ even for the simplest case of two-agent, two-strategy load balancing games. There we prove that such dynamics can provably lead to limit cycles or even chaotic behavior.

For a small enough learning rate $\epsilon$ the behavior of $MWU_e$ approaches that of its smooth variant, replicator dynamics, and hence convergence is once again guaranteed [29]. This means that as we increase the learning rate $\epsilon$ from near zero values we start off with a convergent system and we end up with a chaotic one. Numerical experiments establish that between the convergent region and the chaotic region there exists a range of values for $\epsilon$ for which the system exhibits periodic behavior. Period doubling is known as standard route for 1-dimensional chaos (e.g. logistic map) and is characterized by unexpected regularities such as the Feigenbaum constant [39]. Elucidating these connections is an interesting open problem. More generally, what other type of regularities can be established in these non-equilibrium systems?

Another interesting question has to do with developing a better understanding of the set of conditions that result to non-converging trajectories. So far, it has been critical for our non-convergent examples that the system starts from a symmetric initial condition. Whether such irregular $MWU_e$ trajectories can be constructed for generic initial conditions, possibly in larger congestion games, is not known. Nevertheless, the non-convergent results, despite their non-generic nature are rather useful since they imply that we cannot hope to leverage the power of Baum-Eagon techniques for $MWU_e$. In conclusion, establishing generic (non)convergence results (e.g. for most initial conditions, most congestion games) for $MWU_e$ with constant step size is an interesting future direction.

## Footnotes

[4]$n$ balls $n$ bin games are symmetric load balancing games with $n$ agent and $n$ edges/elements each with a cost function of c(x)=x. We normalize costs equal to $c(x) = x/n$ so that they lie in $[0, 1]$.

[5]This is an absolutely minimal restriction so that the denominator of $\mathrm{MWU}_\ell$ cannot become equal to zero.

[6]A dynamic is called self-independent if the agent's response does not depend on his actions.

[7]All missing proofs can be found in the full version of this paper http://arxiv.org/abs/1703.01138.

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
