[Supplementary Material]

# *Supplementary material* to Multiplicative Weights Update with Constant Step-Size in Congestion Games: Robust Convergence, Limit Cycles and Chaos

## 1 Missing Proofs and material from Section 3

*Proof of Corollary 3.2* . We prove it by doing a reduction. Let $P(\mathbf{x})$ be a non-homogeneous polynomial of degree $d$ on variables $\{x_{ij}\}$ with $\mathbf{x} \in D$ ($D$ is a product of simplices). We introduce a dummy variable $y$ that is always set to one and $D' = \{(\mathbf{x}, y) : \mathbf{x} \in D, y = 1\}$. We define the polynomial $P'(\mathbf{x}, y)$ where for each monomial of $P$ with total degree $d'$ so that $d' \leq d$, we have the same monomial in $P'$ multiplied by $y^{d-d'}$. It is obvious to see that $P'$ is homogeneous of degree $d$. It is also obvious to check that the dynamics as defined in Theorem 2.4 for polynomial $P'$ remains the same as for polynomial $P$ (apart from the extra(dummy) variable $y$ which is always one) since if $y = 1$ at time $t$ then at time $t + 1$, $y$ is equal to $\frac{y \frac{\partial P'(\mathbf{x},y)}{\partial y}}{y \frac{\partial P'(\mathbf{x},y)}{\partial y}} = 1$, *i.e.,* $y$ indeed is always equal to one and $\left. \frac{\partial P'(\mathbf{x},y)}{\partial x_{ij}} \right|_{(\mathbf{x},\mathbf{1})} = \left. \frac{\partial P(\mathbf{x})}{\partial x_{ij}} \right|_{(\mathbf{x})}$.

We conclude that Theorem 2.4 holds for non-homogeneous polynomials. $\qquad\square$

*Proof of Lemma 3.3.* In a congestion game (or potential game; in case of a weighted potential game, the term $\Phi(\mathbf{s})$ below is multiplied by a constant $w_i$), the cost of the function of any player $i$ can be written as the sum of the potential function $\Phi(\mathbf{s})$ and a dummy term which depends on the actions of all the rest players (not on the actions of player $i$), *i.e.,*

$$c_i(\mathbf{s}) = \Phi(\mathbf{s}) + D_i(\mathbf{s}_{-i}). \tag{1}$$

By taking expectations in Equation (1) we get that $\hat{c}_i = \Psi + \mathbb{E}_{\mathbf{s}_{-i} \sim \mathbf{p}_{-i}}[D_i(\mathbf{s}_{-i})]$. Using the law of total expectation it also follows that the expected cost of player $i$ satisfies $\hat{c}_i = \sum_{\gamma \in S_i} p_{i\gamma} c_{i\gamma}$. Therefore $\sum_{\gamma \in S_i} p_{i\gamma} c_{i\gamma} = \Psi(\mathbf{p}) + \mathbb{E}_{\mathbf{s}_{-i} \sim \mathbf{p}_{-i}}[D_i(\mathbf{s}_{-i})]$.

We take the partial derivative of both L.H.S and R.H.S for variable $p_{i\gamma}$ and we conclude that the following holds:

$$c_{i\gamma} = \frac{\partial \Psi(\mathbf{p})}{\partial p_{i\gamma}} + \underbrace{\frac{\partial \mathbb{E}_{\mathbf{s}_{-i} \sim \mathbf{p}_{-i}}[D_i(\mathbf{s}_{-i})]}{\partial p_{i\gamma}}}_{=0}, \text{ thus } \frac{\partial Q(\mathbf{p})}{\partial p_{i\gamma}} = \underbrace{1/\epsilon_i - 1/\beta + 1/\beta \cdot \prod_{j \neq i}\left(\sum_{\gamma \in S_j} p_{j\gamma}\right) - c_{i\gamma}}_{1/\epsilon_i - c_{i\gamma} \text{ since } \mathbf{p} \in \Delta}$$

$$\tag{2}$$

Since the R.H.S of (2) does not depend on $p_{i\gamma}$, $Q$ is a linear function w.r.t $p_{i\gamma}$ for all $i \in \mathcal{N}, \gamma \in S_i$. Therefore, it is a polynomial of degree $N$ with respect to $\mathbf{p}$.

Finally, we will show that all the coefficients of the polynomial $Q$ are non-negative. Let's focus on the monomials containing the term $p_{i\gamma}$ (for some $i, \gamma$). By (2) we have that the summation of those monomials is equal to $(1/\epsilon_i - 1/\beta)p_{i\gamma} + \left(1/\beta \cdot \prod_{j \neq i}\left(\sum_{\gamma \in S_j} p_{j\gamma}\right) - c_{i\gamma}\right) p_{i\gamma}$ which expands to $(1/\epsilon_i - 1/\beta)p_{i\gamma} + \left(1/\beta \cdot \sum_{\mathbf{s}_{-i} \in \mathbf{S}_{-i}} \prod_{j \neq i} p_{j\mathbf{s}_j} - c_{i\gamma}\right) p_{i\gamma}$, where $\mathbf{S}_{-i} \overset{\text{def}}{=} \times_{j \neq i} S_j$. However, we have

$$c_{i\gamma} = \sum_{\mathbf{s}_{-i} \in \mathbf{S}_{-i}} \prod_{j \neq i} p_{j\mathbf{s}_j} \cdot \underbrace{\left( \sum_{e \in \gamma} c_e \left( 1 + k_e(\mathbf{s}_{-i}) \right) \right)}_{\leq \frac{1}{\beta} \text{ by definition of } \beta},$$

where $k_e(\mathbf{s}_{-i})$ denotes the number of players apart from $i$ that choose edge $e$ in the strategy profile $\mathbf{s}_{-i}$. Combining everything together we have that summation of all monomials including $p_{i\gamma}$ is equal to:

$$(1/\epsilon_i - 1/\beta)p_{i\gamma} + \left( 1/\beta - \underbrace{\left( \sum_{e \in \gamma} c_e(1 + k_e(\mathbf{s}_{-i})) \right)}_{\leq \frac{1}{\beta}} \right) \cdot \sum_{\mathbf{s}_{-i} \in \mathbf{S}_{-i}} \prod_{j \neq i} p_{j\mathbf{s}_j} \cdot p_{i\gamma}$$

Clearly, each summand has a nonnegative coefficient. Hence, each monomial containing $p_{i\gamma}$ has a nonnegative coefficient. The above is true for all $i, \gamma$ and the claim follows. $\qquad\square$

*Proof of Theorem 3.4.* By Lemma 3.3, $Q(\mathbf{p})$ is a polynomial with nonnegative coefficients. Therefore, we can apply Corollary 3.2 for polynomial $Q$. In this case, the Baum-Eagon theorem defines the map:

$$p_{i\gamma}(t+1) = \left( p_{i\gamma}(t) \frac{\partial Q}{\partial p_{i\gamma}} \bigg|_{(\mathbf{p}(t))} \right) \bigg/ \sum_{\delta \in S_i} p_{i\delta} \frac{\partial Q}{\partial p_{i\delta}} \bigg|_{(\mathbf{p}(t))}$$

$$\overset{(2)}{=} \frac{p_{i\gamma}(t)(1/\epsilon_i - c_{i\gamma})}{\sum_{\delta \in S_i} p_{i\delta}(t)(1/\epsilon_i - c_{i\delta})} = p_{i\gamma}(t) \frac{1/\epsilon_i - c_{i\gamma}}{1/\epsilon_i - \hat{c}_i},$$

which coincides with MWU$_\ell$ (1). Thus, it is true that $Q(\mathbf{p}(t+1)) > Q(\mathbf{p}(t))$ unless $\mathbf{p}(t+1) = \mathbf{p}(t)$. This proof justifies the reason we added the term $\sum_{i \in \mathcal{N}} \left( (1/\epsilon_i - 1/\beta) \cdot \sum_{\gamma \in S_i} p_{i\gamma} \right) + 1/\beta \cdot \prod_{i \in \mathcal{N}} \left( \sum_{\gamma \in S_i} p_{i\gamma} \right)$ in $Q$, namely so that the partial derivatives give us MWU$_\ell$ dynamics. $\qquad\square$

*Proof of Corollary 3.6.* Let $n$ be the number of players. Given the sets $S_t, S_t'$ and the game $G$, we define a new game $G'$ with $|S_t|$ players (the players of game $G'$ are copies of the players in $S_t$ of game $G$). For each edge $e$ with cost function $c_e$, we introduce a new set of edges $e_0, ..., e_{n-|S_t|}$ where the cost of $c_{e_i}(l) = \Pr[\text{exactly } i \text{ of } n - |S_t| \text{ use edge } e]c_e(l+i)$. It is easy to check that the game $G'$ is still a congestion game and has the same potential as the original game $G$. The rest follows from Theorem 3.1. Observe that it might be the case that the original game is not in a Nash equilibrium (therefore fixed point for MWU$_\ell$), whereas the "subgame" $G'$ is in a Nash equilibrium (i.e., no player in $S_t$ decreases his cost by deviating). In words, the potential is not necessarily *strictly* decreasing but decreasing. As long as at least one player deviates, then it is strictly decreasing. $\qquad\square$

*Proof of Theorem 3.7.* Let $\Omega \subset \Delta$ be the set of limit points of an orbit $\mathbf{p}(t)$. $\Psi(\mathbf{p}(t))$ is decreasing with respect to time $t$ by Theorem 3.1 and so, because $\Psi$ is bounded on $\Delta$, $\Psi(\mathbf{p}(t))$ converges as $t \to \infty$ to $\Psi^* = \inf_t \{\Psi(\mathbf{p}(t))\}$. By continuity of $\Psi$ we get that $\Psi(\mathbf{y}) = \lim_{t \to \infty} \Psi(\mathbf{p}(t)) = \Psi^*$ for all $\mathbf{y} \in \Omega$. So $\Psi$ is constant on $\Omega$. Also $\mathbf{y}(t) = \lim_{n \to \infty} \mathbf{p}(t_n + t)$ as $n \to \infty$ for some sequence of times $\{t_i\}$ and so $\mathbf{y}(t)$ lies in $\Omega$, i.e. $\Omega$ is invariant. Thus, if $\mathbf{y} \equiv \mathbf{y}(0) \in \Omega$ the orbit $\mathbf{y}(t)$ lies in $\Omega$ and so $\Psi(\mathbf{y}(t)) = \Psi^*$ on the orbit. But $\Psi$ is strictly decreasing except on equilibrium orbits and so $\Omega$ consists entirely of fixed points. $\qquad\square$

## 2 Missing Proofs and material from Section 4

In this section, we prove the existence of limit cycles as well as Li-Yorke chaos for MWU$_e$ in the simple congestion games with two agents that have been defined in section 3. To improve readability, we present these examples below.

We consider a symmetric two agent congestion game with two edges $e_1, e_2$. Both agents have the same two available strategies $\gamma_1 = \{e_1\}$ and $\gamma_2 = \{e_2\}$. We denote $x, y$ the probability that the first and the second agent respectively choose strategy $\gamma_1$.

For the first example, we assume that $c_{e_1}(l) = \frac{1}{2} \cdot l$ and $c_{e_2}(l) = \frac{1}{2} \cdot l$. Computing the expected costs we get that $c_{1\gamma_1} = \frac{1+y}{2}$, $c_{1\gamma_2} = \frac{2-y}{2}$, $c_{2\gamma_1} = \frac{1+x}{2}$, $c_{2\gamma_2} = \frac{2-x}{2}$. $\text{MWU}_e$ then becomes $x_{t+1} =$

$x_t \dfrac{(1-\epsilon_1)^{\frac{(y_t+1)}{2}}}{x_t(1-\epsilon_1)^{\frac{y_t+1}{2}}+(1-x_t)(1-\epsilon_1)^{\frac{2-y_t}{2}}}$ (first player) and $y_{t+1} = y_t \dfrac{(1-\epsilon_2)^{\frac{x_t+1}{2}}}{y_t(1-\epsilon_2)^{\frac{x_t+1}{2}}+(1-y_t)(1-\epsilon_2)^{\frac{2-x_t}{2}}}$ (second player). We assume that $\epsilon_1 = \epsilon_2$ and also that $x_0 = y_0$ (players start with the same mixed strategy. Due to symmetry, it follows that $x_t = y_t$ for all $t \in \mathbb{N}$, thus it suffices to keep track only of one variable (we have reduced the number of variables of the update rule of the dynamics to one) and the dynamics becomes $x_{t+1} = x_t \dfrac{(1-\epsilon)^{\frac{x_t+1}{2}}}{x_t(1-\epsilon)^{\frac{x_t+1}{2}}+(1-x_t)(1-\epsilon)^{\frac{2-x_t}{2}}}$. Finally, we choose $\epsilon = 1 - e^{-10}$ and we get

$$x_{t+1} = H(x_t) = x_t \frac{e^{-5(x_t+1)}}{x_t e^{-5(x_t+1)} + (1-x_t)e^{-5(2-x_t)}},$$

i.e., we denote $H(x) = \frac{xe^{-5(x+1)}}{xe^{-5(x+1)}+(1-x)e^{-5(2-x)}}$.

For the second example, we assume that $c_{e_1}(l) = \frac{1}{4} \cdot l$ and $c_{e_2}(l) = \frac{1.4}{4} \cdot l$. Computing the expected costs we get that $c_{1\gamma_1} = \frac{1+y}{4}$, $c_{1\gamma_2} = \frac{1.4(2-y)}{4}$, $c_{2\gamma_1} = \frac{1+x}{4}$, $c_{2\gamma_2} = \frac{1.4(2-x)}{4}$.

$\text{MWU}_e$ then becomes $x_{t+1} = x_t \dfrac{(1-\epsilon_1)^{\frac{(y_t+1)}{4}}}{x_t(1-\epsilon_1)^{\frac{y_t+1}{4}}+(1-x_t)(1-\epsilon_1)^{\frac{1.4(2-y_t)}{4}}}$ (first player) and $y_{t+1} =$

$y_t \dfrac{(1-\epsilon_2)^{\frac{x_t+1}{4}}}{y_t(1-\epsilon_2)^{\frac{x_t+1}{4}}+(1-y_t)(1-\epsilon_2)^{\frac{1.4(2-x_t)}{4}}}$ (second player). We assume that $\epsilon_1 = \epsilon_2$ and also that $x_0 = y_0$ (players start with the same mixed strategy. Similarly, due to symmetry, it follows that $x_t = y_t$ for all $t \in \mathbb{N}$, thus it suffices to keep track only of one variable and the dynamics becomes $x_{t+1} = x_t \dfrac{(1-\epsilon)^{\frac{x_t+1}{4}}}{x_t(1-\epsilon)^{\frac{x_t+1}{4}}+(1-x_t)(1-\epsilon)^{\frac{1.4(2-x_t)}{4}}}$. Finally, we choose $\epsilon = 1 - e^{-40}$ and we get

$$x_{t+1} = G(x_t) = x_t \frac{e^{-10(x_t+1)}}{x_t e^{-10(x_t+1)} + (1-x_t)e^{-14(2-x_t)}},$$

i.e., we denote $G(x) = \frac{xe^{-10(x+1)}}{xe^{-10(x+1)}+(1-x)e^{-14(2-x)}}$.

## 2.1 Analyzing $x_{t+1} = H(x_t)$

**The signs of the derivative of $H(H(x))$**

In this subsection we analyze the monotonicity of $H(H(x))$.

**Lemma 1.** *There exist numbers $0 < y_0 < x_0 < 1/2 < x_1 < y_1 < 1$ so that:*

- *For $x \in [0, y_0], [x_0, x_1]$ and $[y_1, 1]$ $H(H(x))$ is strictly increasing,*

- *for $x \in [y_0, x_0]$ and $x \in [x_1, y_1]$ $H(H(x))$ is strictly decreasing,*

*where $x_0 = \frac{1}{10}(5 - \sqrt{15}) \approx 0.1127$, $x_1 = \frac{1}{10}(5 + \sqrt{15}) \approx 0.8873$, $y_0 \in (0, x_0)$ so that $H(y_0) = x_0$ and $y_1 \in (x_1, 1)$ so that $H(y_1) = x_1$.*

*Proof.* First of all it holds that $\frac{dH(H(x))}{dx} = H'(H(x)) \cdot H'(x)$, therefore we will analyze the signs of $H'(H(x))$ and $H'(x)$ separately. Direct calculations give $H'(x) = e^{5+10x} \frac{1-10x+10x^2}{(e^{10x}(-1+x)-e^{5x})^2}$. The roots of $1 - 10x + 10x^2$ are $x_0$ and $x_1$ (defined in the statement). We conclude that $H$ is strictly increasing in $[0, x_0]$ and $[x_1, 1]$ and strictly decreasing in $[x_0, x_1]$.

Moreover $H(x_0) \approx 0.8593 > x_0$ thus lies in $(1/2, x_1)$ and $H(x_1) \approx 0.1406 < x_1$ and hence lies in $(x_0, 1/2)$. Let $y_0 \in (0, x_0)$ so that $H(y_0) = x_0$ (since $H$ is strictly increasing in $[0, x_0]$, $H(0) = 0$ and $H(x_0) > x_0$, there exists a unique $y_0$) and by similar argument let $y_1$ the unique real in $[x_1, 1]$ so that $H(y_1) = x_1$.

Figure 1: Detailed plot of $H^2$.

We have the following cases:

- For $x \in (0, y_0)$ we get that both $H'(x)$ and $H'(H(x))$ are positive and hence $H(H(x))$ is strictly increasing in $[0, y_0]$ (area 1 of the figure 1).

- For $x \in (y_0, x_0)$ we get that $H'(x)$ is positive and $H'(H(x))$ is negative, thus $H(H(x))$ strictly decreasing in $[y_0, x_0]$ (area 2 of the figure 1).

- For $x \in (x_0, x_1)$ we get that $H'$ is negative and since $(H(x_1), H(x_0)) \subset (x_0, x_1)$, H is monotone we have that $H'(H(x))$ is also negative, namely $H(H(x))$ is strictly increasing in $[x_0, x_1]$ (areas 3,4,5 and 6 of the figure 1).

- For $x \in (x_1, y_1)$ we get that $H'(x)$ is positive and $H'(H(x))$ is negative and hence $H(H(x))$ is strictly decreasing in $[x_1, y_1]$ (area 7 of the figure 1).

- For $x \in (y_1, 1)$ we get that $H'(x)$ is positive and $H'(H(x))$ is positive, thus $H(H(x))$ strictly increasing in $[y_1, 1]$ (area 8 of the figure 1).

$\square$

**The fixed points of $H(H(x))$**

**Lemma 2.** *$H(H(x))$ has 5 fixed points, $0 < \rho_1 < 1/2 < \rho_2 = 1 - \rho_1 < 1$. Moreover $H(H(x)) - x$ is positive in $(0, \rho_1)$, $(1/2, \rho_2)$ and negative in $(\rho_1, 1/2)$, $(\rho_2, 1)$.*

(a) Exponential $\text{MWU}_e$: Plot of function $H$ (blue) and its iterated versions $H^2$ (red), $H^3$ (yellow). Function $y(x) = x$ is also included.

(b) Linear $\text{MWU}_\ell$: Plot of function $H_\ell$ (blue) and its iterated versions $H_\ell^2$ (red) and $H_\ell^3$ (yellow). Function $y(x) = x$ is also included.

*Proof.* By direct calculations we get that

$$H(H(x)) = \frac{x}{\left(e^{-5+10x}(1-x)+x\right)\left(\frac{x}{e^{-5+10x}(1-x)+x} + e^{-5+\frac{10x}{e^{-5+10x}(1-x)+x}}\left(1 - \frac{x}{e^{-5+10x}(1-x)+x}\right)\right)}$$

$$= \frac{x}{x + e^{10x\left(1+\frac{1}{e^{-5+10x}(1-x)+x}\right)-10}(1-x)}$$

It is clear that $H(H(0)) = 0$, $H(H(1)) = 1$ and $H(H(1/2)) = 1/2$. In order to find the other fixed points, it suffices to analyze the roots of the function $1 - x - e^{10x\left(1+\frac{1}{e^{-5+10x}(1-x)+x}\right)-10}(1-x)$. By cancelling the common factor $(1 - x)$ (we have already take into account $x = 1$), we have to analyze $g(x) \stackrel{\text{def}}{=} 1 - e^{10x\left(1+\frac{1}{e^{-5+10x}(1-x)+x}\right)-10}$. It follows by the monotonicity of $e^x$ that $g(x) = 0$ iff $10x\left(1 + \frac{1}{e^{-5+10x}(1-x)+x}\right) - 10 = 0$, i.e., $\frac{x}{e^{-5+10x}(1-x)+x} = 1 - x$.

To solve the equation above, it suffices to analyze the roots of the function

$$g_1(x) \stackrel{\text{def}}{=} x - (1-x)\left(e^{-5+10x}(1-x) + x\right) = x^2 - e^{-5+10x}(1-x)^2.$$

By direct calculation we have to find the roots of $g_2(x) \stackrel{\text{def}}{=} x - e^{-2.5+5x}(1 - x)$ (since $0 \le x \le 1$). Finally, we take the derivative of $g_2$ which is $g_2'(x) = 1 + e^{-2.5+5x} - 5e^{-2.5+5x}(1 - x) = 1 + e^{-2.5+5x}(5x - 4)$. Clearly $g_2''(x)$ is negative in $[0, 3/5)$, positive in $(3/5, 1]$ and zero at $3/5$. Also $g_2'(0) \approx 0.67 > 0$, $g_2'(3/5) \approx -0.648 < 0$ and $g_2'(1) > 0$, i.e., by Bolzano's theorem $g_2'(x)$ has a unique root in $(0, 3/5)$ (say $\alpha_1$) and a unique root in $(3/5, 1)$ (say $\alpha_2$). Finally, since $g_2'(1/2) = -0.5 < 0$ and $g_2'(x_0) \approx 0.504 > 0$, it follows that $x_0 < \alpha_1 < 1/2$ and since $g_2'(x_1) \approx 4.026$ we get that $1/2 < \alpha_2 < x_1$. By the above and Rolle's theorem we conclude that $H(H(x))$ has at most 3 distinct fixed points apart from $0, 1$. Since $g_2$ is increasing in $(0, x_0)$ and $g_2(x_0) \approx -0.015 < 0$, $g_2$ has no root in $(0, x_0]$. Moreover, since $g_2(1/4) \approx 0.035 > 0$, it follows that $g_2$ has a root in $(x_0, 1/4)$ (say $\rho_1$). Hence $H(H(\rho_1)) = \rho_1$ and $1/2 > 1/4 > \rho_1 > x_0$. By observing that $H(1 - x) = 1 - H(x)$, we get that $H(1 - H(x)) = 1 - H(H(x))$ and also $H(H(1 - x)) = H(1 - H(x))$, i.e.,

$$H(H(1 - x)) = 1 - H(H(x)).$$

We substitute $x$ with $\rho_1$ and we get $H(H(1 - \rho_1)) = 1 - H(H(\rho_1)) = 1 - \rho_1$, namely $\rho_2 \stackrel{\text{def}}{=} 1 - \rho_1 > 3/4$ is the remaining fixed point of $H(H(x))$. Whether $H(H(x)) - x$ is positive or negative follows by same arguments. See also the figure 1 for a visualization of this theorem. $\square$

**Periodic orbits**

*Proof of Theorem 4.1.* Since $(\rho_1, 1/2) \subset [x_0, x_1]$, from Lemma 1 it holds that $H(H(x))$ is strictly increasing in $(\rho_1, 1/2)$. Thus if $\rho_1 < x < 1/2$, it follows $\rho_1 = H(H(\rho_1)) < H(H(x)) <$

(c) Exponential $\text{MWU}_e$: Plot of function $H^{10}$. Function $y(x) = x$ is also included.

(d) Linear $\text{MWU}_\ell$: Plot of function $H_\ell^{10}$. Function $y(x) = x$ is also included.

Figure 2: We compare and contrast $\text{MWU}_e$ (left) and $\text{MWU}_\ell$ (right) in the same two agent two strategy/edges congestion game with $c_{e_1}(l) = \frac{1}{2} \cdot l$ and $c_{e_2}(l) = \frac{1}{2} \cdot l$ and same learning rate $\epsilon = 1 - e^{-10}$. $\text{MWU}_e$ converges to a limit cycle whereas $\text{MWU}_\ell$ equilibrates. Function $y(x) = x$ is also included in the graphs to help identify fixed points and periodic points.

$H(H(1/2)) = 1/2$, i.e., the interval $[\rho_1, 1/2]$ is invariant under $H \circ H$. Consider an initial condition $z_0 \in (\rho_1, 1/2)$ and define the sequence $z_{i+1} = H(H(z_i))$. It is clear that $z_i \in (\rho_1, 1/2)$ for all $i \in \mathbb{N}$ from previous argument. Additionally, $(z_i)_{i \in \mathbb{N}}$ is strictly decreasing because $z_{i+1} = H(H(z_i)) < z_i$ (from Lemma 2 we have $H(H(x)) < x$ for all $x \in (\rho_1, 1/2)$). Finally, $z_i > \rho_1$ for all $i \in \mathbb{N}$ (lower bounded), and thus the sequence converges to some limit $l$. It is easy to see that $\rho_1 \leq l < 1/2$ and also $H(H(l)) = l$ by continuity of $H$, namely $l = \rho_1$ (using Lemma 2). Therefore, we showed that for any initial point $z_0 \in [\rho_1, 1/2)$, we get that $\lim_{t \to \infty} H^{2t}(z_0) = \rho_1$. Analogously holds that for any initial point $z_0 \in (1/2, \rho_2]$, we get that $\lim_{t \to \infty} H^{2t}(z_0) = \rho_2$. It is clear that $\lim_{t \to \infty} H^{2t}(1/2) = 1/2$ ($1/2$ is a fixed point of $H$).

Moreover a point $z \in (x_0, \rho_1)$ we have that $z' = H(H(z)) \in (HH(x_0), HH(\rho_1))$ ($H \circ H$ is strictly increasing by Lemma 1). Since $z < \rho_1$, we have that $z' = H(H(z)) > z$ (from Lemma 2). Therefore for any initial point $z_0 \in (x_0, \rho_1)$, the sequence $(H^{2t}(z_0))_{t \in \mathbb{N}}$ is strictly increasing and bounded by $\rho_1$, hence it converges. By similar argument as before we conclude that $\lim_{t \to \infty} H^{2t}(z_0) = \rho_1$. Analogously, it holds for any initial point $z_0 \in (\rho_2, x_1)$ that $\lim_{t \to \infty} H^{2t}(z_0) = \rho_1$.

We continue by considering the case that $z \in (y_0, x_0)$. From Lemma 1 we have that $z' = H(H(z)) \in (H(H(x_0)), H(H(y_0)))$. From Lemma 2 $H(H(x_0)) > x_0$ and $H(H(y_0)) = H(x_0) < x_1$. Therefore $z' \in (x_0, x_1)$ and from the previous cases we have that $\lim_{t \to \infty} H^{2t}(z) = \rho_1$ or $\rho_2$, unless $z' = 1/2$, i.e., unless $H(H(z)) = 1/2$. It is completely analogous the case $z \in (x_1, y_1)$.

To finish the proof, assume $z_0 \in (0, y_0)$. From Lemma 1 is holds that $z_1 = H(H(z_0)) > z_0$. Let $n$ be the minimum index for $t$ so that $z_n = H^{2n}(z_0) > y_0$ ($n$ exists and is finite, otherwise the sequence $(H^{2t})_{t \in \mathbb{N}}$ would converge to a fixed point, which is contradiction because there is no fixed point in $(0, y_0)$). It is clear that $z_{n-1} < y_0$ and hence

$$y_0 < H(H(z_{n-1})) < H(H(y_0)) = H(x_0) < x_1.$$

So either $z_n = 1/2$ or $H(H(z_n)) = 1/2$ or else the sequence $H^{2t}$ converges to $\rho_1$ or $\rho_2$ (by reduction to the previous cases). Completely analogous is the remaining case $z_0 \in (y_1, 1)$.

Therefore we showed the following: For all $z \in (0, 1)$, either there exists a number $k \in \mathbb{N}$ so that $H^{2k}(z) = \frac{1}{2}$ or the limit $\lim_{t \to \infty} H^{2t}(z)$ exists and is equal to $\rho_1$ or $\rho_2$. Finally, the set $\{z \in (0, 1) : \exists k \in \mathbb{N} \text{ s.t } H^{2k}(z) = \frac{1}{2}\}$ has measure zero (from Lemma 1, the set $\{z : H(H(z)) = 1/2\}$ has cardinality at most 5). See also figure 2(c) for a visualization of the theorem. In contrast, figure 2(d) shows that the linear variant converges to the fixed point $1/2$ ($x = 1/2, y = 1/2$ is a Nash equilibrium of the corresponding game, i.e., the first example of Section 4). □

## 2.2 Analyzing $x_{t+1} = G(x_t)$

**Lemma 3.** *G has 3 fixed points $0 < 3/4 < 1$ in $[0, 1]$.*

(a) Exponential MWU$_e$: Plot of function $G$ (blue) and its iterated versions $G^2$ (red), $G^3$ (yellow). Function $y(x) = x$ is also included.

(b) Linear MWU$_\ell$: Plot of function $G_\ell$ (blue) and its iterated versions $G_\ell^2$ (red) and $G_\ell^3$ (yellow). Function $y(x) = x$ is also included.

*Proof.* Let $x$ be a fixed point of $G$. If $x \neq 0, 1$ then $1 + x = \frac{14}{10}(2 - x)$, therefore $x = \frac{3}{4}$. $\qquad\square$

**Lemma 4.** *There exist a $y \in [0, 1]$ so that $G(G(G(y))) = y$, $G(y) \neq y$, $G(G(y)) \neq y$ and $G(G(y)) \neq G(y)$. Hence $y, G(y), G(G(y))$ is a periodic orbit of length three.*

*Proof.* It holds that $G(G(G(0.4))) - 0.4 \approx -0.158$ and $G(G(G(0.5))) - 0.5 \approx 0.496$ and hence by Bolzano's theorem there exists a $y \in (0.4, 0.5)$ so that $G(G(G(y))) = y$. Observe that $y$ cannot be a fixed point of $G$ because of Lemma 3. If $G(G(y)) = y$, then by applying $G$ we get $G(G(G(y))) = G(y)$ and hence $y = G(y)$ (contradiction since $y$ cannot be a fixed point). Finally, if $G(G(y)) = G(y)$ then by applying $G \circ G$ we get $G(G(G(G(y)))) = G(G(G(y)))$, and since $G(G(G(y))) = y$ we have that $G(y) = y$ (contradiction again). See also figure 3(a) for a visualization of the theorem. $\qquad\square$

Using Li-Yorke theorem and Lemma 4 we can show Theorem 4.2.

*Proof of Theorem 4.2.* It follows from Li-Yorke theorem (Theorem 2.3) and Lemma 4. See also figure 3(c) for a visualization of the theorem. In contrast, figure 3(d) shows that the linear variant converges to the fixed point $3/4$ ($x = 3/4, y = 3/4$ is a Nash equilibrium of the corresponding game, i.e., the second example of Section 4). $\qquad\square$

Finally using Theorem 4.2 we show that for any $1 > \epsilon > 0$, we can create games so that MWU$_e$ exhibits chaotic behavior for infinitely many initial conditions.

*Proof of Corollary 4.3.* Given any $1 > \epsilon > 0$ and $n$, consider a game with 2 edges $e_1, e_2$ and a dummy edge that does not belong to the strategy set of players $n - 1, n$. Assume that the costs for the two edges are $c_{e_1}(x) = al$ and $c_{e_2}(l) = bl$ where $a = \frac{10}{\ln 1/(1-\epsilon)}$ and $b = \frac{14}{\ln 1/(1-\epsilon)}$. The first $1, 2, ..., n - 2$ players choose the dummy edge with probability one. MWU$_e$ dynamics ensures that the $n - 2$ players don't change their strategy along the iterations of the dynamics (if a strategy is played with probability zero, that probability remains zero for all times). For players $n - 1, n$, let $x, y$ be the probabilities to choose edge $e_1$ and we start from the symmetric position $x = y$. It is not hard to show that the update rule of the MWU$_e$ dynamics is $\frac{x(1-\epsilon)^{a(1+x)}}{x(1-\epsilon)^{a(1+x)}+(1-x)(1-\epsilon)^{a(2-x)}} = \frac{xe^{-10(1+x)}}{xe^{-10(1+x)}+(1-x)e^{-14(2-x)}}$, namely the same as $G(x)$ for both players, i.e., we reduce the instance to that of our second example and by Theorem 4.2 our claim follows. $\qquad\square$

(c) Exponential $\mathrm{MWU}_e$: Plot of function $G^{10}$. (d) Linear $\mathrm{MWU}_\ell$: Plot of function $G_\ell^{10}$. Function
Function $y(x) = x$ is also included.                        $y(x) = x$ is also included.

Figure 3: We compare and contrast $\mathrm{MWU}_e$ (left) and $\mathrm{MWU}_\ell$ (right) in the same two agent two strategy/edges congestion game with $c_{e_1}(l) = \frac{1}{4} \cdot l$ and $c_{e_2}(l) = \frac{1.4}{4} \cdot l$ and same learning rate $\epsilon = 1 - e^{-40}$. $\mathrm{MWU}_e$ exhibits sensitivity to initial conditions whereas $\mathrm{MWU}_\ell$ equilibrates. Function $y(x) = x$ is also included in the graphs to help identify fixed points and periodic points.