[Reviews · NeurIPS 2017]

Reviewer 1



The paper studies the dynamics associated with the multiplicative weights meta algorithm in the context of congestion games. The main question motivating this study is whether applying MWU with constant learning rate yields convergence to exact Nash equilibria. By connecting the dynamics to Baum-Eagon inequality, the authors prove a positive result for the standard MWU algorithm. Interestingly, it is shown that when a different variant of the algorithm is applied with constant step size, it leads to limit cycles or even chaotic behavior. The applicability of this result is questionable; indeed, applying MWU using the doubling trick yields optimal rates and does not incur any overhead. Furthermore, only asymptotic bounds are obtained. However, I still find the contribution interesting and even enlightening. The proofs are elegant and possible connections to other fields such as computational complexity and distributed computing make the contribution even more appealing.

Reviewer 2



The paper revisits the convergence result of multiplicative weighted update (MWU) in a congestion game, due to Kleinberg, Piliouras, and Tardos (STOC 2009), and establishes a connection between MWU and Baum-Welch algorithm in a neat and elegant style. By showing the monotonicity of the potential function in a congestion game, the authors prove that any MWU with linear updating rule converges to the set of fixed points, which is a superset of all the Nash equilibria of the congestion game. The Baum-Eagon inequality offers a new interpretation of the dynamics of the linear updating rule in MWU and their results in congestion games are quite general, and so are the conditions on initialization and isolated Nash equilibrium in a congestion game with finite set of agents and pure strategies. The results in this paper hold for any congestion game irrespective of the topology of the strategy sets by the nature of their game, however, one should emphasize the assumption that, individual earning rates $\epsilon_i$ are bounded above, in order for Baum-Eagon inequality to work in their context. An elaborate analysis on the upper bounds would be nicer, since for any $i$, this paper requires that $$\frac{1}{\epsilon_i} > \sup_{i,\mathbf{p}\in \Delta,\gamma\in S_i} \{c_{i\gamma}\}\geq \max_{e} c_e(|N|),$$ which can be huge when the number of agents is large enough. Although Theorem 3.7 holds, the proof presented by the authors is flawed. To prove any point $\mathbf{y} \in \Omega$ is a fixed point of $\zeta$, one could guess from the argument in the paper that the authors intend to show that $\Phi(\mathbf{y}) = \Phi(\zeta(\mathbf{y}))$, thus by monotonicity of $\Phi$, $\zeta(\mathbf{y})=\mathbf{y}$. However, they applied the notation $\mathbf{y}(t)$, even though $\mathbf{y}$ is, according to their definition, the limit point of an orbit $\mathbf{p}(t)$ as $t$ goes to infinity. The subsequent argument is also hard to follow. A easy way to prove theorem 3.7 is by contradiction. Suppose that $\mathbf{y}$ is not a fixed point, then by definition (line221), there exists some $i,\gamma$ s.t., $y_{i\gamma} > 0$ and $c_{i\gamma}\neq \hat{c}_i$. If $c_{i\gamma} > \hat{c}_i$, then by convergence and continuity of the mapping, $\exists t_0$ and $\epsilon_0 > 0$, s.t., $\forall t > t_0$, $p_{i\gamma}(t) \geq \epsilon_0 > 0$, and $c_{i\gamma}(t) \geq \hat{c}_i(t)+ \epsilon_0.$ Then it's not hard to see that $$p_{i\gamma}(t+1) = p_{i\gamma}(t) \frac{\frac{1}{\epsilon_i} - c_{i\gamma}(t) }{\frac{1}{\epsilon_i} - \hat{c}_{i}(t)} \leq p_{i\gamma}(t) (1- \epsilon_0 \epsilon_i),$$ which yields $\lim_{t\to \infty} p_{i\gamma}(t) = y_{i\gamma}=0$, contradicting with $y_{i\gamma} > 0$. If $c_{i\gamma} < \hat{c}_i$, then by convergence and continuity of the mapping, $\exists t_0$ and $\epsilon_0 > 0$, s.t., $\forall t > t_0$, $p_{i\gamma}(t) \geq \epsilon_0 > 0$, and $c_{i\gamma}(t) + \epsilon_0 < \hat{c}_i(t).$ Then it's not hard to see that $$p_{i\gamma}(t+1) = p_{i\gamma}(t) \frac{\frac{1}{\epsilon_i} - c_{i\gamma}(t) }{\frac{1}{\epsilon_i} - \hat{c}_{i}(t)} \geq p_{i\gamma}(t) (1+ \epsilon_0)$$. Then $\lim_{t\to \infty} p_{i\gamma}(t)$ doesn't lie in the simplex. In Remark 3.5, the authors claim that Lemma 3.3 and thus Theorem 3.4 and 3.1 hold in the extension of weighted potential games. However, a underlying step in the proof of Lemma 3.3 might not hold without adding extra conditions: the non-negativity of coefficients in $Q(p)$ in a weighted potential game. Similar to the proof of Lemma 3.3, in the weighted case, $\frac{\partial Q}{\partial p_{i\gamma}} = \frac{1}{\epsilon_i} - \frac{c_{i\gamma}}{w_i}$. Whether the RHS is non-negative is unknown simply by the definition of $\epsilon_i$ nor the fact that $w_i \in [0,1]$. On Line 236, the constant in $Q(p)$ should rather be $Q(p)=const -\Phi$ and $const = \sum_{i\in N} 1/{\epsilon_i} + (|N|-1) 1/\beta$.

Reviewer 3



This paper studies multiplicative weight update algorithm in congestion games. The typical multiplicative weights algorithm updates the probability by an action by 1 - \eps C(\gamma) (linear update) or (1 - \eps)^C(\gamma) (exponential update). In either case, generally the parameter \eps is also updated as the algorithm progresses. In this paper, the authors study what happens when one does not update the parameter \eps. In congestion games, the authors show that the linear update always converges. On the other hand, they construct examples of congestion games where for any values of \eps between (0, 1) the exponential update does not converge but instead exhibits something called Li-Yorke chaos. The paper is very well written and uses novel techniques. Minor nits: * Provide a reference for "it is well known that to achieve sub-linear regret, the learning rate eps must be decreasing as time progresses." Also how does sub-linear regret relate to convergence? * Should Corollary 3.2 be called corollary - I am assuming it generalizes Baum-Eagon but follows using the same/similar proof.